# Force-Generating Mechanism of Axonemal Dynein in Solo and Ensemble

**DOI:** 10.3390/ijms21082843

**Published:** 2020-04-18

**Authors:** Kenta Ishibashi, Hitoshi Sakakibara, Kazuhiro Oiwa

**Affiliations:** 1Graduate School of Frontier Biosciences, Osaka University, Osaka 5650871, Japan; ishiba@nict.go.jp; 2Center for Information and Neural Networks (CiNet), National Institute of Information and Communications Technology, Osaka 5650871, Japan; 3Advanced ICT Research Institute, National Institute of Information and Communications Technology, Kobe 6512492, Japan; sakaki@nict.go.jp; 4Graduate School of Life Science, University of Hyogo, Hyogo 6781297, Japan

**Keywords:** axonemal dynein, axoneme, bend formation, bend propagation

## Abstract

In eukaryotic cilia and flagella, various types of axonemal dyneins orchestrate their distinct functions to generate oscillatory bending of axonemes. The force-generating mechanism of dyneins has recently been well elucidated, mainly in cytoplasmic dyneins, thanks to progress in single-molecule measurements, X-ray crystallography, and advanced electron microscopy. These techniques have shed light on several important questions concerning what conformational changes accompany ATP hydrolysis and whether multiple motor domains are coordinated in the movements of dynein. However, due to the lack of a proper expression system for axonemal dyneins, no atomic coordinates of the entire motor domain of axonemal dynein have been reported. Therefore, a substantial amount of knowledge on the molecular architecture of axonemal dynein has been derived from electron microscopic observations on dynein arms in axonemes or on isolated axonemal dynein molecules. This review describes our current knowledge and perspectives of the force-generating mechanism of axonemal dyneins in solo and in ensemble.

## 1. Introduction

Autonomously orchestrated activities of axonemal dyneins and their interactions with microtubules drive ciliary and flagellar movement. Dynein was first identified in the axonemes of *Tetrahymena* cilia as a high molecular weight protein with Mg-ATPase activity [1,2]. The axonemal dynein is the major component of the dynein arms [2]. In a cross section of the electron micrographs of the axoneme, the dynein arms were observed as two rows of protrusions extending from one side of each doublet microtubule in the axonemes [3]. These are called outer or inner (dynein) arms, depending on their location (Figure 1a). The outer arm of *Chlamydomonas* flagella consists of three distinct dynein heavy chains associated with intermediate and at least 11 distinct light chains. Several components other than dynein are associated with the outer arm [4]. The number of dynein heavy chains in the outer arm depends on the species of origin [5]. As far as investigated, the outer arm of metazoan sperm is comprised of two distinct heavy chains [6,7,8,9], whereas that from *Tetrahymena* [10] and *Chlamydomonas* [11] each contains three distinct heavy chains. The inner arm is not composed of a single dynein species. At least seven dynein subspecies were identified as components of the inner arm in *Chlamydomonas* axonemes [12], arranged along the doublet microtubule with 96-nm longitudinal repeats. The inner arm dynein subspecies I1/f has heterodimeric heavy chains (α and β) and two WD-repeat intermediate chains plus several light chains. The other inner arm dyneins, termed subspecies a, b, c, d, e, and g, consist of monomeric heavy and light chains (actin and centrin/p28) [12,13,14].

These heavy chains of axonemal dyneins so far studied have distinct motile properties in vitro [12,15,16,17,18,19] and specific functions in situ in flagellar motility [20,21]. Mutant analysis of *Chlamydomonas* flagella showed that not all dyneins are necessary for flagellar beating, but certain combinations of dyneins seem to be crucial for the appropriate axoneme function [20]. Furthermore, some heavy chains of axonemal dyneins can generate torque [12,22,23] and can oscillate in solo [24], although the roles of these properties in the axoneme are not yet well understood. For the movement of cilia and flagella, various types of dyneins are required to coordinate to function together.

In the oscillatory bend formation, which has been intensively studied, the active sliding of doublet microtubules is proposed to switch from one side of the axoneme to the other [25,26]. To recognize the active side in the axoneme, we indexed the doublet microtubules according to their positions relative to the plane of the central pair apparatus or according to the presence of the bridge-like structure. From a historical viewpoint, there are two distinct numbering systems. In sea urchin sperms, the bending plane is perpendicular to the plane of the central pair apparatus and bridge-like structure between the pair of doublet microtubules (#5–#6). Thus, the doublet in the bending plane is indexed as #1 and those with bridge-like structures are indexed as #5 and #6 [3]. This indexing has been adopted for the cilia and flagella of many animals, including mammalian spermatozoa [27,28]. Because the central pair apparatus can rotate while beating in *Chlamydomonas*, the bridge structure and doublet microtubule lacking outer arms are used as indexing markers of doublet #1 (the arrowhead in Figure 1a) [29].

The flagellar wave is asymmetric around the long axis of the flagellum, with the degree of bending to one side being substantially greater than that of bending to the other. The bend with the larger curvature is the principal bend (P-bend), and that with the smaller curvature is the reverse bend (R-bend). In *Chlamydomonas*, the effective stroke, in which the flagellum makes an oar-like movement toward one side, is primarily composed of the principal bend and generated mainly using the dynein arms on doublet #3 pulling up doublet #4. The recovery stroke, in which the flagellum moves back by propagating a bend from base to tip, is generated mainly using the dynein arms on doublet #7 pulling up doublet #8 (Figure 1b) [29]. To achieve this alternating episode of activation of dynein arms, the dynein molecules on the doublets must be precisely controlled in space and time. The switching of the active sides seems to be achieved through an antagonistic inhibition of dynein activity.

By contrast to the switching of the active sides, the bend propagation toward the tip of the axoneme seems to be the sequential process activating dynein arms [30,31,32,33]. To understand these coordination mechanisms of axonemal dyneins during ciliary and flagellar beating is one of the ultimate aims for the research on the motility of cilia and flagella. Despite several theoretical and structural studies, however, the mechanisms of the inhibition and the activation among dynein arms to form and propagate bends remain elusive (Reviewed in Lindemann (2010) [34]). In this review, we summarize studies mainly on axonemal dyneins, their force generating mechanisms, and their roles in an axonemal structure.

## 2. Overview of the Dynein Molecule Architecture

Although several atomic-resolution structures of cytoplasmic dynein under different nucleotide conditions have recently been reported [35,36,37,38,39,40], no atomic coordinates of the entire motor domain of axonemal dynein have existed to date. This is because no proper expression system has been established that provides a sufficient biochemically relevant amount of homogeneous axonemal dynein molecules. Under these circumstances, a substantial amount of knowledge for determining the molecular architecture of axonemal dynein has been derived from electron microscopic observations on outer dynein arms in axonemes [41,42,43,44,45,46,47,48,49,50,51] and on isolated axonemal dyneins [52,53,54,55]. In this review, knowledge obtained from the atomic structures of cytoplasmic dynein is generalized because cytoplasmic and axonemal dyneins show high homology of the amino acid sequences in their motor domains [5,56] and are constructed using similar basic components (tail, linker, AAA+ modules, stalk, and buttress/strut) despite their distinct roles in cells.

Compared with myosin and kinesin, which have closely packed domain structures including catalytic and filament-binding domains, dynein has a much larger, clearer modular structure. A dynein heavy chain has a molecular mass of typically 500–540 kDa, consisting of approximately 4500 amino acid residues (Figure 2a). It contains a fundamental motor domain in the C-terminus, which is a fragment of approximately 320–380 kDa that contains sites for both ATP hydrolysis and microtubule binding. Dynein is a member of a protein superfamily known as ATPases associated with diverse activities (AAA+) [57,58]. In most AAA+ proteins, the hexameric ring is formed by six identical AAA+ modules located on separate polypeptide subunits [57]. In contrast to conventional AAA+ proteins, dynein has six AAA+ modules concatenated in a single heavy chain and they form a ring-like motor core with three unique appendages: the linker, stalk, and buttress/strut. Each AAA+ module is typically composed of a large α/β subdomain (large domain) containing the nucleotide binding Walker A and B motifs and a small α-helical subdomain (small domain) extending from the C-terminus of the large domain. The large and small domains form the top layer of the linker faces and the C-terminal face of the head ring, respectively (Figure 2b,c).

The first four AAA+ modules (AAA1–AAA4) contain a highly conserved Walker A motif (a so-called P-loop) and a Walker B motif [59,60,61,62]. As clearly shown by the dynein crystal in the ADP state, all of these modules can bind nucleotides [39]. By contrast, AAA5 and AAA6 have highly degraded Walker motifs. The Walker A and B motifs of AAA1 are the principal sites of ATP hydrolysis [63]. Other Walker motifs found in AAA2−4 bind either ADP or ATP and function in regulatory roles. In cytoplasmic dyneins, mutations in the AAA3 ATPase site impair their motility so severely that the Walker motif in AAA3 is thought to play important roles in motility [63,64,65]. Some axonemal dyneins require the presence of ADP for motility in vitro and increase the velocity of microtubules in the presence of ADP [66,67,68]. A fluorescent ATP analog was used as a probe of the nucleotide-binding sites of axonemal dynein from sea urchin sperm flagella and showed at least two distinct binding sites [69]. These observations suggest that at least one of these additional AAA+ modules in the axonemal dyneins can bind nucleotides and may play a physiological role in regulation of dynein activity.

### 2.1. Linker

The linker is a part of the tail proximal to AAA1 and functions as a major mechanical element (Figure 2). It changes conformation in a nucleotide-dependent manner, thereby driving microtubules. The linker consists of four helix-rich subdomains (subdomain 1, 2, 3 and 4) and it arches over the head ring (Figure 2b–d). The C-terminal subdomain 4 connects into AAA1 interacting with AAA1 and part of AAA6. N-terminal subdomain 1 contacts AAA5 in the apo state [35], but AAA4 in the ADP state [39,40]. In addition to those in the AAA5 large domain, patches that are enriched in conserved residues also exist in the large domain of AAA1 and in two β-hairpins of AAA2. These patches of conserved residues stabilize the position of the linker in the docking sites on the AAA+ ring [37].

Both the motor domains in the apo state and in the ADP state have a straight linker (Figure 2c) while the motor domain in the ADP-vanadate (Vi)-state (mimicking the ADP-Pi state) has a linker bending between subdomains 2 and 3 at the central linker cleft (Figure 2b,d) [38]. Thus, the linker can be divided into static and mobile subdomains. The hinge helix spans the central linker cleft between static and mobile subdomains. Mobile subdomains 1 and 2 move like a rigid-body relative to the static subdomains 3 and 4. Subdomain 2 interacts with the AAA2 site that forms a highly conserved hydrophobic patch of amino-acids. The hydrophobic interaction between AAA2 and subdomain 2 forces the hinge linker to adopt a curved conformation [38] storing recoil energy.

### 2.2. Stalk and Microtubule-Binding Domain

The stalk is the coiled-coil extension of the small domain helices of AAA4 and is composed of the CC1-helix outgoing from and the CC2-helix incoming into AAA4 [35,40]. A globular microtubule-binding domain (MTBD) is at the tip of the stalk. Binding of ATP to the primary ATPase site of AAA1 facilitates the dissociation of MTBD from the microtubule, and binding of the MTBD to the microtubule accelerates the release of hydrolysis products from the ATPase site. Because the site of the MTBD spatially separates from the primary ATP hydrolysis site, finding the communication pathway between them has been an important concern for understanding dynein function. Based on several studies, it was hypothesized that the shear between CC1-helix and CC2-helix modulates the structure of MTBD and influences its affinity to microtubules [36,70,71]. As mentioned below, the discovery of the buttress/strut has provided evidence that the change in the interaction between the stalk helices and the buttress/strut coupled with nucleotide states elicits inter-strand sliding of the stalk and subsequent conformational changes in the MTBD [38].

Recent cryo-electron microscopic observations on the MTBD-microtubule complex have shown that MTBD binding to the microtubule induces distortion in the microtubule lattice [72]. When the microtubules were fully decorated with the MTBDs of axonemal dynein DNAH7, the microtubule cross-sectional curvature was largely distorted, but the distortion was not observed when the microtubule was decorated by the MTBD of cytoplasmic dynein-1. The authors claimed that the dynein coordination in an axoneme could be mediated through conformational changes in the microtubule as is found in microtubules fully decorated with the MTBD.

### 2.3. Buttress/Strut

The buttress [35]/strut [40] is an anti-parallel coiled-coil extension emerging from the AAA5 small domain and bending sharply at its middle. It extends toward the stalk and connects at its tip, forming a Y-shaped structure together with the stalk. Because this Y-shaped structure had been previously observed using electron microscopy on axonemal dynein subspecies c [52], the buttress/strut is probably a common architecture in all dynein types.

Atomic coordinates of dynein in the ADP state [39] and in the ADP-Vi states [38] show the change in the interaction between the stalk and the buttress/strut. This change provides insight into the signal pathway between MTBD and AAA1. In the dynein-ADP structure, the two helices of the stalk are in a straight conformation at the stalk/buttress interface, while in the dynein-ADP-Vi structure, movement of the buttress/strut causes a bulging of CC2-helix at the stalk-buttress/strut contact site. This bulging induces the relative sliding of CC2-helix with respect to CC1-helix and changes the affinity of the MTBD to a microtubule. Thus, the buttress/strut relays rigid body motion between AAA+ modules (especially the rotation of the AAA6 large domain and the AAA5 small domain) into the shear between the α-helices of the stalk (Figure 2b).

### 2.4. Tail

The tail serves as a platform for multimerization of dynein heavy chains and for the assembly of different intermediate chains and light intermediate chains (for cytoplasmic dynein) to form the dynein complex [73]. Because expressed cytoplasmic dyneins without their native tail but with a substituted artificial tail can move processively in vitro [74,75], these subunits are not essential for dynein motility in vitro. However, recent data indicate that the tail is involved in linking dynein to cargo and to several adaptor proteins as well as in regulating dynein motility in vivo [73].

In cytoplasmic dyneins, the tail is involved in auto-inhibition [75]. This inhibition involves two heads interacting with each other to form a phi (φ)-shaped configuration. The phi-shaped dynein can interact with a microtubule through a microtubule-binding domain at the tip of the stalk, but it then moves diffusively on the microtubule. Disrupting the phi-shaped configuration by inserting a small obstacle between two heads or by connecting two heads with a rigid rod increases the affinity of dynein to microtubules and, as a result, the dynein molecule then moves processively on the microtubule. For axonemal dyneins, the autoinhibition configuration plays a role in the transport of axonemal dyneins in intra-flagellar transport (IFT) [76,77].

In an axoneme, the tail mediates binding of associated subunits of dynein and anchors the heavy chain onto the A-microtubule of one doublet microtubule. In *Chlamydomonas* axonemal dyneins, monomeric inner arm dyneins (named subspecies a, b, c, d, e, and g) have a complement of associated proteins, including monomeric actin and either an essential light chain (p28 for a, c, and d) [12,78,79] or the calcium-binding calmodulin homolog centrin (for b, e, and g) [12,80]. These light chains are thought to be associated with the tail domain. The tail domain of the outer arm dynein is involved in the interaction with docking complex components which are required for dynein assembly at the correct location within the axoneme [81]. These docking complexes contain Ca^2+^ binding subunits, DC3, which may help to coordinate dyneins in response to the change in Ca^2+^ concentrations. The linkage network between dynein heavy chains, light chains, and intermediate chains are reviewed by King (2016) [4].

## 3. Force Generating Mechanism of Dynein

At the essence of force-generation of dynein molecules is the contraction of the molecule by swing motion of the mobile part of the linker coupled with ATP hydrolysis [52]. Two-dimensional electron microscopy showed that the tail emerges near the base of the stalk in the absence of a nucleotide (post-power stroke conformation). In the presence of orthovanadate (VO_3_^−4^, Vi), a dynein ATPase inhibitor acting as a phosphate analog, dynein is trapped in the ADP-Pi state (pre-power stroke conformation), and the emergence point of the tail moves further away from the stalk base. These two-dimensional observations have been interpreted as the swing motion of the linker [52]. Cryo-electron microscopy [54] and X-ray crystallography of dyneins [82,83] have revealed the detail of the linker movement as described previously.

As mentioned above, electron microscopy and X-ray crystallography have provided invaluable information on the static structures of dynein. However, it is still necessary to investigate the linker movement during the ATP hydrolysis cycle under dynamic and more physiological conditions. Furthermore, the relationships between the linker motions and biochemical intermediate states of the ATP hydrolysis cycle should be determined to reveal the mechanism underlying dynein motility. Kon and his colleagues detected dynamic conformational changes of the linker by using Förster resonance energy transfer (FRET) measurements [84]. A green fluorescent protein was attached to the N-terminal end of the linker and a blue fluorescent protein (BFP) was inserted into the head ring of the *Dictyostelium* cytoplasmic dynein. The linker movement relative to the head ring was detected as the change in FRET efficiency. This FRET analysis of the resultant motor domain provides strong evidence that dynein adopts at least two functional states (states I and II) and the tail undergoes ATP-induced motions relative to the head domain during transition between the two states. This enabled us to assign the snapshots of dynein structures obtained using X-ray crystallography and electron microscopy to these two functional states (Figure 3, [85]).

When viewed from the linker side, the head ring of apo dynein arranges AAA+ modules with uneven spacing. Large gaps appear between the large domains of AAA1 and AAA2, as well as between those of AAA5 and AAA6. In the ADP state, the gap between AAA5 and AAA6 is not clearly observed, and the gap between AAA1 and AAA2 is partially filled by the linker (Figure 2c). In addition, the dynein motor domain in the ADP state is more symmetric and planer than that in the apo state. The distortion of the head ring coupled with the presence or absence of bound nucleotide represents the signal pathway between the MTBD of the stalk and the principal ATPase site in AAA1.

Atomic coordinates of dynein in the ADP-vanadate state filled a blank remaining in the mechanochemical cycle of dynein (Figure 3). Binding of ATP into the ATPase site (in the gap between AAA1 and AAA2) begins closing the gap and to compress the head ring (Figure 3b). The gap closure induces a shift in the position of AAA5, which changes the interaction between the stalk and buttress/strut. The buttress/strut then pulls the CC2-helix of the stalk, resulting in shear between the CC1-helix and CC2-helix in the stalk, causing MTBD to release the microtubule (Figure 3c). Compression of the head ring induces a steric crash of the linker against the head ring, and this crash forces the linker to adopt a kinked configuration (Figure 3d) [38]. Hydrolysis of ATP increases the AAA1-2 gap and induces the rotational shift in the position of the AAA5 large domain. The shift of AAA5 pushes the buttress/strut toward the AAA4 domain and the buttress/strut–stalk interaction is restored to its former state, increasing the affinity of MTBD for microtubule. The MTBD binding to the microtubule leads to a further increase in the AAA1–2 gap, facilitating the release of hydrolysis products. Upon the release of these products, the linker changes its orientation while releasing the energy stored in the hinge helix of the linker.

## 4. Force Generation of Dynein in the Axoneme

The questions of how axonemal dynein generates force in situ and how these conformational changes in the motor domain occur in the axoneme have taken years of study to elucidate. Early electron micrographs of axonemes described the outer arm dynein as electron-dense blobs with a hook in the cross-section [3,86], as three stacked spheres [87,88], or a hammer [89] in the longitudinal view. The limited spatial resolution of electron microscopy at that time and low accessibility to dynein arms in situ resulted in confusion of the nucleotide-dependent conformational changes of dynein arms. However, the advent of cryo-electron tomography has enabled great progress for understanding structure and conformational changes of axonemal dynein at much higher resolutions [46,50,51,90,91,92]. Tomograms of the axoneme reveal that three heavy chains of outer arm dyneins (α, β, and γ) stack on the top each other, and four such trios align with the 96-nm periodicity. Eight heavy chains of inner arm dyneins (a, b, c, d, e, fα, fβ, and g) appear to form four pairs: fβ–fα, a–b, c–e, and g–d (Figure 4).

Global changes of the dynein arms were found when cryo-electron tomograms of axonemes in the apo state (post-power stroke state) were compared with those in the ADP-vanadate state (pre-power stroke state). Although the stalks typically tilt toward the proximal end (flagellar base) of the axoneme in both nucleotide conditions [47], the dynein head rings [93] in the apo state shifted 8 nm toward the distal end of the axoneme, compared with the position of the head in the ADP-vanadate state (Figure 4f). An electron density map of the inner arm dynein subspecies c obtained as an isolated molecule fit well to an averaged tomogram of the axoneme if the head and stalk are rotated [54]. This suggests that the conformational change observed in isolated dynein molecule occurs even in situ. Furthermore, high resolution cryo-electron tomography on sea urchin sperm flagella clearly succeeded in imaging the previously unresolved linker and stalk, thus showing the rotation of the head ring relative to the linker [49]. Under this tension (force generation), the dynein molecule would be stretched, and the flexible parts (the stalk and the linker neck) and the dynein head domain are aligned. If the head domain rotates, the molecule would contract as a result and pull the adjacent doublet microtubule [93,94].

Dynein conformation changes in motion were first captured by cryo-electron microscopy [95]. The tail truncated *Dictyostelium* cytoplasmic dynein was artificially dimerized to move stepwise along a microtubule. This dynein was then rapidly frozen during its movement. The image obtained is thought to be that of a dynein molecule in action. Single-particle image processing reveals that both dynein stalks point toward the microtubule minus end. The stalk angle of the rear head becomes shallower while the stalk angle of the front head becomes steeper when the distance between both microtubule binding domains increases. These observations suggest that the dynein head rotates and conducts a Brownian search for the next binding site by extension of the stalk. Thus, once the binding domain attaches to a binding site, the head ring begins rotation and pulls the head ring toward the minus end. Because the head rings of the outer arm dynein in *Chlamydomonas* flagella have been shown to stack like coins (Figure 4a) [46], the movement of the head rings during their sliding could resemble the cytoplasmic dynein in action.

## 5. In the Axoneme: Conversion of the Shear between Doublets into the Bend

The shear generated by the movement of the dynein head is converted into the bend in the axoneme. Experimental evidence of this conversion was provided by Shingyoji and her colleagues [96]. A demembranated sea urchin sperm flagellum was captured with a fine glass needle in ATP-free reactivating solution and was reactivated locally by iontophoretic application of a small amount of Mg-ATP. The local sliding induced by the applied ATP generated typical S-shaped bends, in which the straight region was flanked by the equal bends with opposite curvatures.

Autonomous bend propagation along the axoneme also has been shown experimentally [97,98,99]. When a flagellum losing coordination and ceasing to beat was mechanically bent near the base with a microneedle, the flagellum restarted propagation of the bend. In this propagation of the bend toward the tip of the flagellar axoneme, the mechanically activated region propagated toward the tip of the axoneme [32]. Therefore, bend propagation seems to be the activation process of the dynein arms adjacent to the bend region (Figure 5a,b).

For continuous beating, however, a mechanism must exist that antagonistically switches on and off the activity of dynein arms on each side of the axoneme [25,26]. Dyneins on one side of the axoneme are actively working during principal bend formation, and those on the opposite side must be inhibited or relaxed, and vice versa during the reverse bend formation. Numerous experimental and computational results support the “switch point” hypothesis predicting that dyneins on one side of the axoneme will be in a different functional state from those on the opposite side within a single axonemal bend (reviewed by Lindemann (2010) [34]). The control mechanism would then reside within the axoneme because isolated and demembranated axonemes are capable of autonomous functioning.

Two major concepts are proposed to explain the mechanisms of switching the activity of the dynein arms. (i) Dynein arms detect the signal of the switching point autonomously through interaction with microtubules. (ii) Additional components other than dynein arms play the primary role of regulation of dynein activities (e.g., [102,103]). In the latter case, the most-widely known concept of the coordination is control through the interactions between the central pair apparatus and the radial spoke complexes as several lines of experimental evidence have been reported using flagellar mutants of *Chlamydomonas*. For instance, Oda and his colleagues [104] showed that a motility defect in a central pair projection mutant was rescued by the addition of exogenous protein tags on the radial spoke heads. This result suggests that dynein activity is regulated by the mechanical signals between the central pair apparatus and the radial spokes. Another candidate for mechanical trigger of dynein activities is the spoke tilting. A spoke-central-pair apparatus interaction may trigger the enzymatic regulatory cascade of kinases residing on the spoke, propagating signals to the inner arm dyneins. However, a recent publication by Yagi and his colleagues clearly showed that *pf14* (radial spoke missing mutant) and *pf18* (central pair less mutant) flagella can beat the wave form under ultra-high pressure [105]. Their paper strongly suggests that the beating of the flagellum can be generated only by the interaction between dynein and microtubules.

Switching activity of the dynein arms occurs twice per beat cycle. In *Chlamydomonas* flagella, the rate is above 100 times per second. This rapid coordination is reasonably thought to result from mechanical feedback. Some models propose that a regulation mechanism resides in the dynein arms themselves through interaction with microtubules. At least three different mechanisms describing how dynein detects the switching point have been proposed, although these three are not mutually exclusive.

(i) A sliding control model [106,107,108,109] posits that in the antagonistic pair of dynein arm arrays, sliding force generated by dyneins on one side builds up a load that accelerates dyneins on the other side to detach from the microtubule. The consequence of this tug-of-war is a catastrophic detachment of dyneins on one side of the axoneme, leading to an imbalance of sliding forces and therefore to axonemal bending. Stable oscillation in this model requires elastic components.

(ii) The curvature control model [33,110,111] states that the system automatically introduces a quarter-cycle phase shift because the active process that caused the rate of curvature change was activated by the passive bending resulting from the internal shear near an abrupt change in curvature. Therefore, rhythmic bending waves are automatically generated by a control process, in which active sliding is a function of the flagellum curvature [110]. Recently, the curvature control model has been updated. Dynein arms are proposed to detect alterations in the rate of change in axonemal curvature [112].

(iii) The geometric clutch model [113,114,115] proposes that detachment of dynein is regulated by transverse forces acting to separate adjacent doublet microtubules when they are curved. The transverse force becomes larger than the binding force of dynein to the microtubule, so the dynein is detached from the microtubule, and sustained tension rapidly decreases.

Experimental verification of any of these models has not yet been well achieved. Therefore, it is unclear as yet which, if any, of these mechanisms regulates the beating of the axoneme. As indicated in Figure 5c, specific dynein subsets are perhaps activated during travel of the bend [116]. However, these models do not assign any particular conformational states to dyneins within straight or curved regions. Experimental observations are expected to be performed to answer what functional states occur and how they change with time.

In electron microscopic studies, the configuration of the dynein arms in various nucleotide states has been described [47,48]. If the coupling between the mechanochemical states of individual dynein arms and the curvature of the axoneme during flagellar beating become elucidated by experimental observations, our understanding of the switching mechanism would greatly progress. Dyneins undergo several structural transitions during a mechanochemical cycle. Recently, cryo-electron tomography with high spatial resolution distinguished these changes in dyneins and revealed the distribution of these dyneins with various mechanochemical states along the sea urchin sperm flagellum [48].

Because each flagellum was rapidly frozen during active beating, it is assumed that the mechano-chemical states of dyneins are preserved in their functional state [48]. In their observations, dyneins in the straight region were in a pre-power stroke conformation. For this observation, this pre-power stroke conformation was thought to yield a force-balanced state in straight regions. Within bends, conversely, dyneins on one side of the axoneme were in an “inhibited” or microtubule-released state, leading to a force imbalance that allows opposing dyneins to undergo a power stroke-causing sliding and thus bending. The authors claim that inhibition of the conformational state and microtubule release on specific doublets would lead to a force imbalance across the axoneme allowing for microtubule sliding and consequently initiation and formation of a ciliary bend. Propagation of this inhibitory signal from base-to-tip and switching the microtubule doublet subsets that are inhibited have been proposed to result in oscillatory motion. The authors claimed that the “switch inhibition” model allows for a robust all-or-none response providing new insight into the mechanisms of oscillatory ciliary beating [48]. These views of the mechanism of bend formation and propagation are different from previously proposed mechanisms. This underscores the importance of understanding the configuration of dynein under force generation.

## 6. Perspectives

Recent technological progress in electron microscopy has now provided not only the molecular organization of single dynein molecules but also the structural basis of the ensemble behavior of dynein molecules in an axoneme. In particular, electron tomography has enabled the analysis of complex and heterogeneous systems, such as the three-dimensional structure of dynein molecules in situ in axonemes. Some observations are, however, in contrast to previous physiological measurements carried out on living cilia and flagella. Therefore, the information provided by electron microscopy should be examined by functional analysis with optical microscopy. To this end, an experimental system is required that functions under geometrically well-defined and mechanically constrained conditions. Such a system should be comparable to the size and the order of the axoneme structure.

DNA origami techniques combined with the single-molecule technique will pave the way to the system for determining the ensemble behavior of dynein molecules [117]. This DNA technique will provide details of the structural basis for the coordination of dynein arm complexes. In addition to the DNA origami techniques, atomic force microscopy will provide the opportunity to observe dyneins in action, in solution and at molecular resolution. If such a system is achieved, we will be able to perform quantitative, reproducible, and precise experiments on individual dynein molecules and ensembles. Dynein molecules may influence each other through stress and strain during coordinated beating and propagation of bending in cilia and flagella. Herein, we strove to provide sufficient background of inner- and outer arm dyneins a concise practical guide to an avenue for functional analysis of the dyneins.

## Figures and Tables

**Figure 1 ijms-21-02843-f001:**
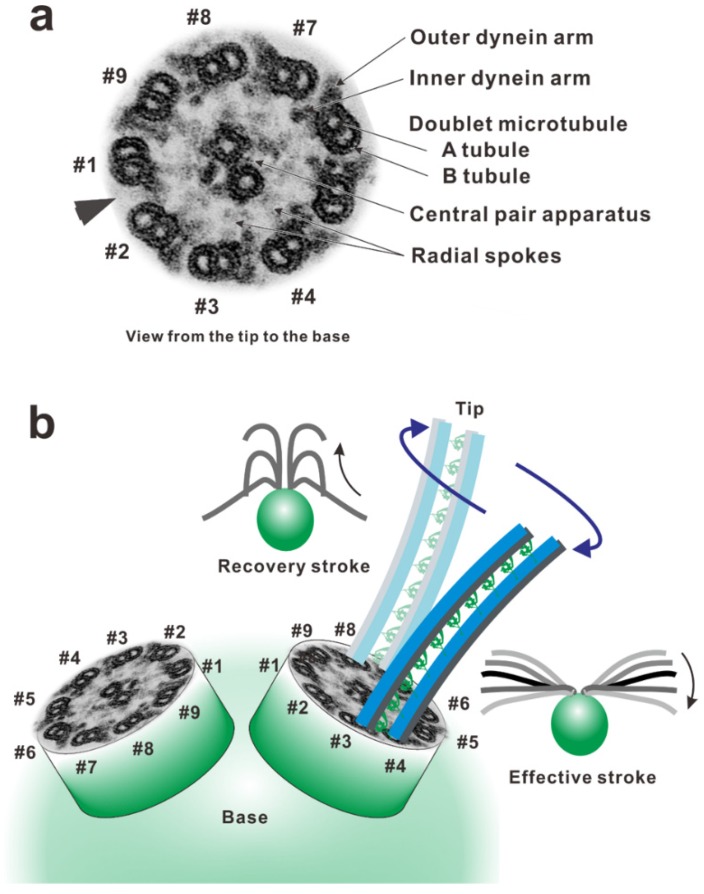
Arrangement of the components in an axoneme. (**a**) A double-stained ultra-thin section electron micrograph of *Chlamydomonas* flagellar axoneme. The arrowhead indicates the lack of the outer arm from the doublet microtubule and the position of the bridge between the doublet. The axoneme was viewed from the distal side to the proximal (to the cell body) (tip-base-view). (**b**) The doublet microtubules were numbered according to the system of Hoops and Witman (1983) [29], based on the flagellar arrangement on a *Chlamydomonas* cell. The doublet microtubule lacking outer dynein arms (the arrowhead in Figure 1a) along the entire length of the axoneme was designated as doublet #1; the remaining doublets were numbered consecutively in a counterclockwise direction, viewing the axoneme from distal to proximal. Doublet #1 of each flagellum faces the other one and is on the outside edge of the principal bend of the effective stroke.

**Figure 2 ijms-21-02843-f002:**
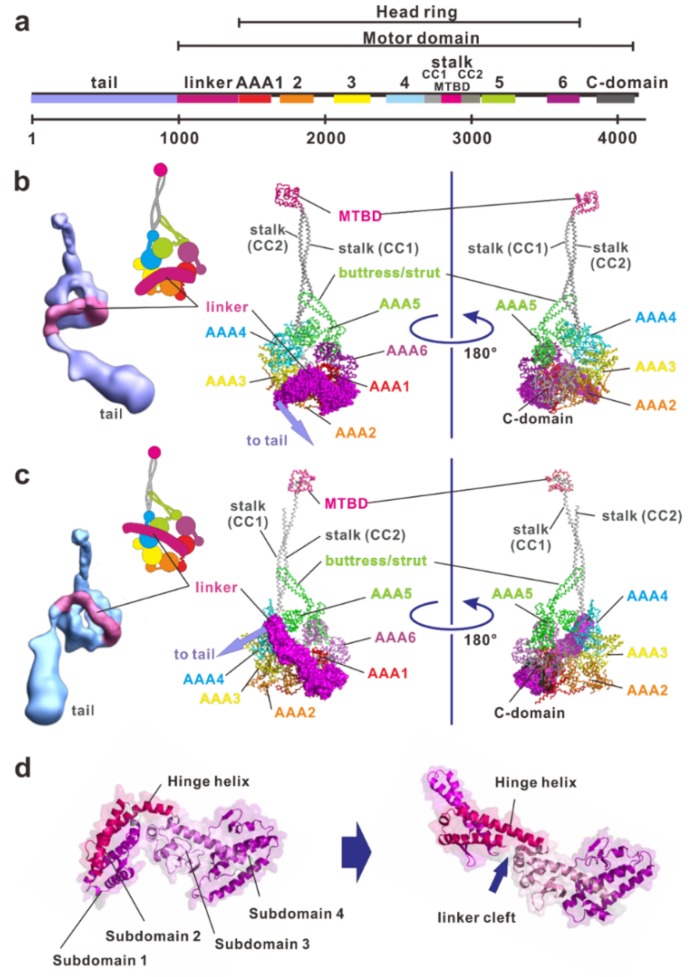
The structure and molecular organization of a dynein. (**a**) Linear map of the heavy chain of *Chlamydomonas* inner arm dynein c (BAE19786). (**b**) The electron density 3D map of the native axonemal dynein c in ADP-vanadate (Vi) state [54] and atomic structure of the dynein-2 motor domain in ADP-Vi state (PDB ID: 4RH7). Concatenated AAA+ modules form a ring, which are indicated in red, orange, yellow, cyan, green, and purple for AAA1 through AAA6, respectively. Inset, a schematic drawing of the dynein heavy chain in the ADP-Vi state. The large and small domains of each AAA+ module are depicted using large and small circles, respectively. (**c**) Electron density 3D map of the whole molecule of the native axonemal dynein c in the apo state (no-nucleotide state) [54]. Atomic structure of the dynein motor domain in the ADP state (PDB ID: 3VKH). A part of the stalk is missing in this structure because of the low electron densities. Inset, a schematic drawing of the dynein heavy chain in the apo state. (**d**) Structural changes of the linker. From N-terminus, subdomain 1, 2, 3, and 4 can be identified. Subdomain 4 is connected into AAA1. In the apo state, the subdomain 1 connects with AAA5, but in the ADP state it connects with with AAA4. The subdomains 1 and 2 are mobile and undergo a rigid-body rotation relative to the static subdomains 3 and 4. These figures were prepared using PyMOL provided by DeLano Scientific LLC (http://www.pymol.org).

**Figure 3 ijms-21-02843-f003:**
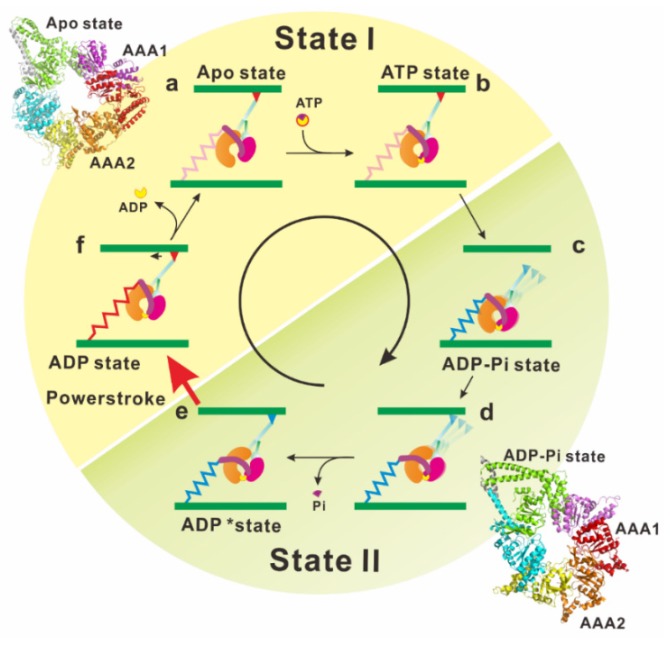
In the apo state, dynein tightly binds to a microtubule via the microtubule-binding domain (MTBD) (**a**). The ATP binding site (the gap between AAA1 and AAA2) then widely opens (see the structure of AAA ring of apo-dynein, PDB ID: 4AKG). The N-terminal end of the linker is docked at a stable position near AAA4/5. Upon binding of Mg-ATP to the primary ATPase site, the gap between AAA1 and AAA2 (AAA1-2 gap) is closed, and a large re-arrangement occurs among AAA+ modules (**b**). This re-arrangement of the AAA+ modules in the head ring changes the interaction between the buttress/strut and stalk. Subsequently, CC1-helix and CC2-helix in the stalk slide past each other, resulting in the MTBD conformational change and subsequent MTBD dissociation from the microtubule (**c**). The linker was sterically clashed with the head ring by the closure of the AAA1-2 gap, and the clash causes the linker to detach from the AAA4/5 docking point and change its orientation on the head ring (see the structure of AAA ring of ADP-Vi-dynein, PDB ID: 4RH7) (**d**). ATP hydrolysis increases the AAA1-2 gap. This change induces the rotational shift in the position of the AAA5 large domain, which pushes the buttress/strut toward the AAA4 domain. Thus, the buttress/strut–stalk interaction is modified and increases the affinity of MTBD to the microtubule (**e**). The MTBD locates and binds the binding site on the microtubule. This binding leads to further increase in the AAA1–2 gap, which releases the hydrolysis products (**f**). Upon the release of the hydrolysis products, the linker orientation on the ring changes, bringing the tail emergence point closer to the stalk. The linker changes its orientation by switching between two differently docked positions on the head ring, thus producing a rotation of the head ring. Reproduced with permission [85]. Copyright 2016 Elsevier, 4782871338086.

**Figure 4 ijms-21-02843-f004:**
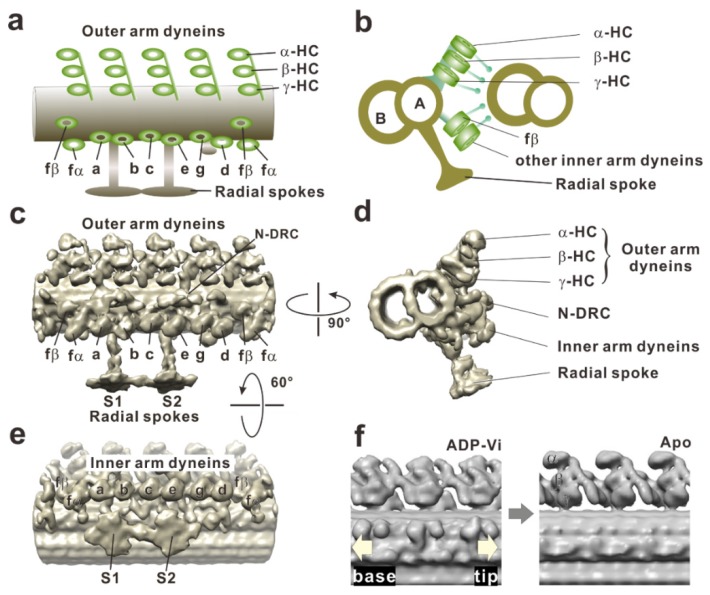
Cryo-electron tomographic images of a doublet microtubule in a *Chlamydomonas* flagellum. (**a**) A schematic drawing of the dynein arms and doublet microtubule in the longitudinal section. Three heavy chains of outer arm dyneins (α, β, and γ) stack on top of each other, and four such trios align with the 96-nm periodicity. The left and right sides indicate proximal (base) and distal (tip) sides, respectively. (**b**) A schematic drawing of the doublet microtubules in the cross-section viewed from the flagellar base to the tip. (**c**) Surface-rendered representations of a doublet microtubule in a longitudinal section on the basis of cryo-electron tomography (EMD-2116). Dynein arms arrayed on the doublet microtubule are viewed from the adjacent doublet microtubule. A structure connects two adjacent doublet microtubules is the nexin-dynein regulatory complex (N-DRC). There are two radial spokes (S1 and S2). (**d)** Cross-section viewed from the proximal (from the cell body) to the distal side. The image clearly shows the three rows: outer dynein arm, inner dynein arm, and N-DRC. (**e**) Longitudinal section of the doublet microtubule viewed from the central pair apparatus. The arrangement of the inner arm dyneins is clearly visible. Eight heavy chains of inner arm dyneins (a, b, c, d, e, fα, fβ, and g) appear to form four pairs: fα–fβ, a–b, c–e, and g–d. (**f**) Conformational changes of outer arm dyneins coupled with nucleotide states were observed in the comparison of two tomography images, EMD-1696 and EMD-1697, from Movassagh et al. (2010) [47].

**Figure 5 ijms-21-02843-f005:**
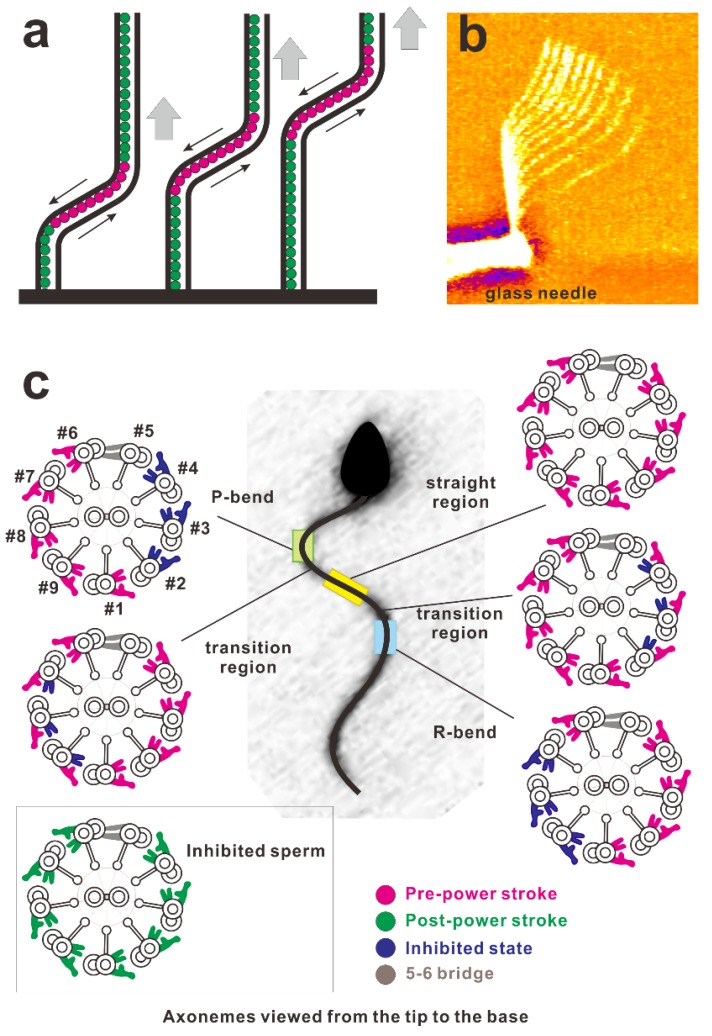
(**a**) Schematic of the propagation of a bend in the axoneme. Red circles indicate the active dynein arms generating shear between the microtubules and green ones indicate inactive dynein arms. A bend activates dynein arms and propagates toward the tip. (**b**) Overlaid video images of a demembranated flagellum spontaneously propagating the bend applied near the base using the rapid movement of the glass needle (courtesy of Dr. Shingyoji). Soaking a demembranated flagellum in low concentrations of Mg-ATP (e.g., 2.5 μM) ceases its beating. Rapid movement of the microneedle generates a bend of the flagellum at the region near the base. This bend propagates toward the tip with a constant bend angle [32]. (**c**) Asymmetric distributions of dynein arms in the active state coupled with the bend revealed by electron tomography. (This diagram was redrawn with modifications according to Lin and Nicastro (2018) [48].) The sea urchin sperm flagellum during its swimming was rapidly frozen and observed using cryo-electron tomography. In this figure, the cross-section of the axoneme was drawn as the view from the distal (flagellar tip) to proximal (cell body) ends of the flagellum. The configuration of dynein molecules was categorized into four states and the distribution was examined using the tomography results. For further details of the observations, consult Lin et al. (2018) [48], Hastie et al. (1991) [100], or King (2018) [101]. Asymmetric distributions of the dynein states are shown to be coupled with bends.

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
