# Peer review of "Force-Generating Mechanism of Axonemal Dynein in Solo and Ensemble"

_ijms, 2020, doi:10.3390/ijms21082843_

Round 1

Reviewer 1 Report

The goal of the submitted review entitled “Force-generating mechanism of axonemal dynein in 2 solo and ensemble” by Oiwa and co-authors was to summarize the current knowledge and present the perspectives in the understanding of the mechanism(s) behind the activity of the axonemal dyneins.

Looking from the perspective of a student or a newcomer to the field, someone who has only limited knowledge about the axonemal dyneins, the first part (pages 1-4) of this review is confusing (see details below), and should be read by an English native speaker, while the latter part is well written.

Authors used different terms to describe dyneins and this is confusing (do the Authors refer to the entire complex – a dynein arm or to the motor protein (dynein heavy chain, a subunit of the dynein arm)?. In an Abstract there are: axonemal dynein, dyneins, dynein arms and axonemal dynein molecules. In the introduction there is the following statement :

 “These dynein molecules are organized with a few heavy chains that form heterotrimers, heterodimers, or monomers, together with intermediate chains (IC), light intermediate chains (LIC) and light chains (LC)” this suggests that “dynein molecules” mean “dynein arm”.

It seems that the term “molecule” was used before to describe a single motor protein (a dynein heavy chain). Based on the sentence cited above, it is still hard to say, what are the components of the outer and inner dynein arms.

Few lines below, the Authors mention that in the metazoan sperm, the ODAs are two-headed while in Tetrahymena and Chlamydomonas are three-headed. What about other cilia in metazoan e.g. tracheal or ependymal cilia in humans? Doe all ciliated or flagellated single-cell organisms have three-headed ODAs or it is a group-specific ODAs composition?

The Authors stated that: “In this review, knowledge obtained from the atomic structures of cytoplasmic dynein is generalized because cytoplasmic and axonemal dyneins are constructed with similar basic components, despite their distinct roles in cells.”

It is true that to some extent, the description of the dynein heavy chains can be generalized. However, detailed analyses of the cytoplasmic dynein-1 and IFT motor, dynein-2 showed some differences, e.g.in the consensus of the Walker A and B sequence in AAA2, AAA3 and AAA4 (Roberts, Biochemical Society Transactions (2018) 46 967–982, https://doi.org/10.1042/BST20170568). When Authors describe Walker motifs in AAA domains in dynein heavy chains, do they generalize or provide data concerning only axonemal dyneins? Do all axonemal dyneins (subunits of ODAs and IDAs) has the same consensus in the Walker motifs?

And more general suggestion (based on this example) – it would be really helpful to indicate which information is “generalized” and which were obtained based on the axonemal dynein heavy chain analyses.

Fig.2A. The linear map of the heavy chain of Chlamydomonas inner-arm dynein c. The cyan was used to mark “tail” while light blue is used to mark AAA4. In the schematic drawing of the dynein heavy chain in apo state AAA4 is in cyan while the tail is missing (?) (please compare to Box1 in the Roberts, 2018).

The term “apo state” is used without explanation what does it mean. Please define. It would be helpful to first introduce all dynein states (apo state, ADP state , ADP-Pi state, DP-Vi-state).

Line 120: “In addition to those in the AAA5 large domain, patches that are enriched in conserved residues..” What do Authors mean by “patches”?

CC1-helix (please add coiled-coil)

Fig 5 – all IDAs are shown as two-headed. In cilia 6 out of the 7 IDAs are single-headed.

Minor comments

Not all REF are cited: In Introduction – first few lines the cited REF are: 2, 3, 7, 12. But REF1, 4, 5, 6, 8-11 are missing.

Frequently a space between the word and [REF] is missing e.g. line 30 (is: …activity[2, 3], should be:….activity [2, 3].).

Papers are cited separately:

Is:…. and a Walker B motif [51],[52],[53],[54].”  Should be: ….and a Walker B motif [51-54].

Line 33:

Is: These dynein molecules are organized with a few heavy chains..

suggested change: These dynein arms are composed with few heavy chains

Fig. 1. I would suggest omitting the expression ”double staining”. The double staining of the TEM ultrathin sections with uranyl and lead salts is a routine technique.

Reviewer 2 Report

This is a very detail and important paper descrying the mechanism of axonemal dynein, which would be useful for a general audience.

Unfortunately, the paper is hard to understand by a non-expert. Adding original figures, clarifying the present figures, and adding explicit definition to used terms will help to make the subject matter be better understood. 

Comments

“In the oscillatory bend formation” – pleas add what the other modes are? How are they different from each other?

“In Chlamydomonas, the effective stroke is generated 49 mainly by the dynein arms on doublet #3 pulling up doublet #4, and the recovery stroke is generated 50 mainly by the dynein arms on doublet #7 pulling up doublet #8” – is this different in Human? If yes, what is the situation there? 

“The switching of the active sides seems to be achieved through 53 an antagonistic inhibition of dynein activity” – add a reference

Figure 2 font is too small

“In contrast to conventional AAA+ proteins, however, six 88 AAA+ modules are concatenated in a heavy chain and form a ring-like motor core with three unique 89 appendages, the linker, stalk and buttress/strut.” - In contrast? What is the normal case?

“a large 90 α/β subdomain (large domain) and small α-helical subdomain (small domain) (Figure 2).” – please add the “large domain” and “small domain” notation to figure 2

“The Walker A and B motifs of AAA1 are the principal sites of ATP 107 hydrolysis” - add a reference

The paper uses jargon that is not familiar to the general audience. Please, define (maybe in a text box): effective stroke, recovery stroke, principal bend, apo state, rigid-body rotation 

 “The linker consists of four helix-rich subdomains (subdomain 1, 2, 3 and 4) and it arches 117 over the head ring (Figure 2d)” should be (Figure 2a or 2C)

Please keep the color of the linker the same in the various parts of figure 2.

“Based on several studies, it was 139 hypothesized that the shear between CC1 and CC2 modulates the structure of MTBD and influences 140 its affinity to microtubules [31, 61, 62].” -Please mark CC1 and CC2 in figure 2.

Is “CC1 and CC2 modulates” the same as “CC1-helix” and “CC2-helix” – keep the same name to reduce confusion.

In figure 2, please indicate the side of the dined that is stably bound to the microtubules and the side that lesebinds during struck.

Add notation of the tail in figure 2

“(post-power stroke conformation, state I)” add “Figure 2.”

The terms “state I” and “state II” are confusing – why not to use post-power stroke conformation and power stroke conformation. I suggest eliminating the terms state I and state II from text and figures.

Figure 2 – add the structure and color code in the figure including AAA5, CC1, CC2, and MTBD, and the rest

Please define the distal end of the axoneme – also add to figures

- “The shear generated by the movement of the dynein head is converted into the bend in the axoneme.” – p explain the centriole role.

- Fig 1 and 5 are from different spices - Please explain if doublet numbers are similar and if not then how they differ

- Fig 5 - define P-Band and R-bend

Reviewer 3 Report

In this manuscript, Ishibashi and his colleagues reviewed the mechanical force generating molecular mechanism of dynein motor proteins in flagella and motile cilia axonemes. Axonemal dyneins in the form of inner and outer arms that associated with microtubule doublets provide the motive force for the movement of flagella by hydrolysis ATP. Recent cryo-EM studies suggest that dynein protein undergoes conformational change upon ATP hydrolysis which affects its affinity to microtubules. This confirmation change drives the movement of microtubules. Overall, this is a well-organized paper with proper and sufficient summarization. I have some suggestions on the figure presentations.

Suggestions:

  1. Figure 2: The ring like motor core structure formed by six AAA+ modules is not very clear due to the small size of the atomic structure and the presence of the linker domain. Zoom in this core structure only to show the arrangements of these 6 modules without the linker domain will help reader to understand the gap between module 1 and 2 are larger than others.
  2. The tail domain is discussed in the main text, but its position is not indicated in the models in Figure 2.
  3. Fiugre 4. There are too many domains in cryo-EM surface rendered images. For general readers without deep knowledge about cryo-EM, showing models is much easier to understand than showing raw experimental data in a review manuscript.
  4. Line 211, is bigins a typo of begins?

Round 2

Reviewer 1 Report

A manuscript was significantly improved ! It is a very nice and interesting review that will greatly contribute to the field.

Very minor poins.

Introduction – page 1 – space is missing

“… depends on the species of origin[5].”

Introduction – page 1

“… The inner arm is not composed..:

Suggested change “.. The inner arms are not composed..” (Authors refer to all 7 IDAs)